# GeoSketch: A Neural-Symbolic Approach to Geometric Multimodal Reasoning with Auxiliary Line Construction and Affine Transformation

## Abstract

Geometric Problem Solving (GPS) poses a unique challenge for Multimodal Large Language Models (MLLMs), requiring not only the joint interpretation of text and diagrams but also iterative visuospatial reasoning. While existing approaches process diagrams as static images, they lack the capacity for dynamic manipulation—a core aspect of human geometric reasoning involving auxiliary line construction and affine transformations. We present GeoSketch, a neural-symbolic framework that recasts geometric reasoning as an interactive perception–reasoning–action loop. GeoSketch integrates: (1) a Perception module that abstracts diagrams into structured logic forms, (2) a Symbolic Reasoning module that applies geometric theorems to decide the next deductive step, and (3) a Sketch Action module that executes operations such as drawing auxiliary lines or applying transformations, thereby updating the diagram in a closed loop. To train this agent, we develop a two-stage pipeline: supervised fine-tuning on 2,000 symbolic-curated trajectories followed by reinforcement learning with dense, symbolic rewards to enhance robustness and strategic exploration. To evaluate this paradigm, we introduce the GeoSketch Benchmark, a high-quality set of 390 geometry problems requiring auxiliary construction or affine transformations. Experiments on strong MLLM baselines demonstrate that GeoSketch significantly improves stepwise reasoning accuracy and problem-solving success over static perception methods. By unifying hierarchical decision-making, executable visual actions, and symbolic verification, GeoSketch advances multimodal reasoning from static interpretation to dynamic, verifiable interaction, establishing a new foundation for solving complex visuospatial problems.

## 1 Introduction

With the advent of Multimodal Large Language Models (MLLMs) (OpenAI, 2024; Comanici et al., 2025; Hong et al., 2025), Geometric Problem Solving (GPS) presents a unique challenge to MLLMs, demanding not only the joint understanding of text and diagrams but also rigorous, multi-step deductive reasoning (Zhang et al., 2023; Qiao et al., 2024; He et al., 2025). While modern MLLMs can ingest multimodal inputs, their reasoning output remains confined to static text. This limits the use of dynamic and visuospatial strategies in geometric problem solving, which becomes particularly evident in complex tasks requiring multi-stage manipulation.

Human geometric problem-solving is inherently interactive and dynamic (Christou et al., 2005; Freksa et al., 2019). As shown in Figure 1, confronted with a complex proof, experts do not merely analyze a static diagram; they actively manipulate it through a temporal sequence of actions. This process involves auxiliary line construction to reveal latent properties and affine transformations—including reflection, rotation, and translation—to reconfigure components, establish congruences, and deduce new relationships. Each action is a deliberate step in a logical argument, creating a visual proof that evolves over time. This synergy of symbolic logic and spatial manipulation remains a critical gap in AI capabilities (Kuang et al., 2025; Wang et al., 2025b; Yan et al., 2025; Cho et al., 2025).

As the field of multimodal reasoning evolving, it has been moving beyond static visual perception towards a "think with image" cognitive workspace (Su et al., 2025). In GPS, the challenge lies in encoding this step-by-step reasoning process that mirrors how humans intuitively manipulate geometric objects to

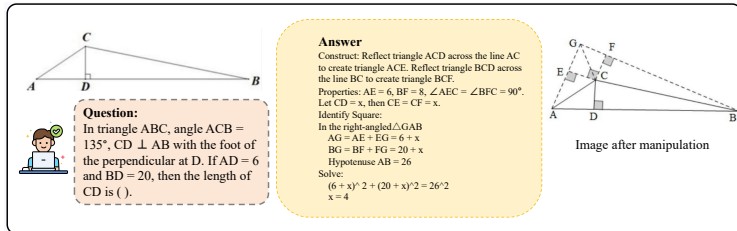

Figure 1: An visual geometric problem that require dynamic image manipulation

solve problems. Recent advancements, such as Visual Sketchpad (Hu et al., 2025), have introduced image-based reasoning through sketching. Yet, these approaches predominantly treat sketching as a monolithic final answer, failing to model the executable, iterative reasoning process that leads to it. True geometric reasoning requires a dynamic workspace where logical thought and spatial action are intertwined.

To bridge this gap, we introduce GeoSketch, a neuro-symbolic framework that recasts geometric problem-solving as a structured, multimodal interaction loop. GeoSketch enables dynamic, ordered manipulation of the diagram, allowing an agent to incrementally build a solution through a synergy of symbolic logic and visual execution. Our framework is built on a hierarchical three-tier architecture that mirrors the human problem-solving process: 1) **Visual Perception**: Processes the input diagram using visual tools (e.g., YOLO (Redmon et al., 2015)) to abstract it into a structured, symbolic representation (logic forms). This step distills the pixel-based image into a set of objects and relations that can be rendered losslessly, forming a manipulable digital twin of the diagram. 2) **Symbolic Reasoning**: A theorem-proving module, powered by an instructed MLLM, operates on this symbolic representation. It applies geometric knowledge to deduce the next logical step and decides on a specific, executable sketch operation (e.g., "draw a line from point A to point B" or "rotate triangle ABC by 90 degrees"). 3) **Sketch Action**: A rendering engine executes the commanded operation, altering the diagram. This updated visual state is then fed back into the perception module, closing the loop and enabling the next cycle of perception, reasoning, and action. This architecture ensures that every visual manipulation is grounded in symbolic logic, mitigating hallucination and fostering verifiable reasoning.

To instantiate GeoSketch, we implement it on the Qwen2.5-VL-7B foundation model and adopt a two-stage training strategy. First, we use supervised fine-tuning (SFT) on a high-quality, curated dataset of 2,000 expert trajectories. These trajectories are generated using geometric templates and handcrafted rules to guarantee correctness and diversity, teaching the model valid step-by-step reasoning. Second, we employ reinforcement learning (RL) to allow the model to explore the vast space of valid reasoning paths, enhancing its strategic flexibility and robustness against novel problems. To support and evaluate this new paradigm, we contribute two key resources: first is the 2K GeoSketch fine-tuning dataset, a distilled collection of multi-step visual reasoning trajectories, and second, the GeoSketch benchmark, the largest curated set of 390 geometric problems specifically designed to require auxiliary construction and affine transformations for their solution. Experiment results show that the GeoSketch benchmark challenges current MLLMs, and our architecture boosts the performance compared to former methods.

By unifying hierarchical decision-making, differentiable rendering, and symbolic verification, GeoSketch advances multimodal reasoning from static perception to dynamic, executable interaction. It establishes a new foundation for tackling complex visuospatial problems where the process is as important as the answer.

## 2 RELATED WORK

### 2.1 AUTOMATED GEOMETRY THEOREM PROVING AND PROBLEM SOLVING

Automated geometry theorem proving has evolved from symbolic deduction to multi-modal reasoning. Early systems like iGeoTutor (Wang & Su, 2015) and Inter-GPS (Lu et al., 2021) translate diagrams into formal logic and apply symbolic rules to generate human-readable proofs. By reason-

ing with standard Euclidean axioms rather than complex algebraic proofs, these symbolic systems produce interpretable and verifiable solutions. The symbolic components of GeoSketch follow this line of research as the proofs are both friendly to humans and LLMs.

A significant challenge in this domain is the scarcity of high-quality, structured training data. Geo-DRL (Peng et al., 2023) addressed this by coupling symbolic theorem solvers with a Graph Neural Network (GNN) trained via self-supervised reinforcement learning on geometry logic graphs. Google's Olympiad Solver (Trinh et al., 2024) pushed this paradigm further by generating massive-scale synthetic data to train a specialized transformer model from scratch, enabling it to generate complete proofs conditioned on theorem premises and conclusions. In contrast to these specialized architectures, GeoSketch adopts an agentic training framework built upon a general-purpose Multimodal Large Language Model (MLLM), leveraging its inherent reasoning and multimodal capabilities. UniGeo (Chen et al., 2022) unifies the problem of calculation and proving in the form of sequence generation. Similarly, GeoSketch also unifies different forms of the testing data (multi-choice, gap filling, and proofing) into a gold-labelled answer generation task. Recent work has focused on leveraging large-scale vision-language fine-tuning. GeoGPT4V (Cai et al., 2024) synthesizes basic geometry problems with aligned text and images by leveraging GPT-4 and GPT-4V. Similarly, G-LLAVA (Gao et al., 2025) is trained on 170K geometric image-caption and question-answer pairs to outperform GPT4-V on the geometric task of MathVista (Lu et al., 2024) benchmark. While these models address visual question answering, they operate on a static image representation and lack the capability to dynamically manipulate the diagram, which is a core contribution of GeoSketch. Although neural-symbolic mechanisms have been explored by GeoDRL and Google's Olympiad solver, less work has been done on MLLMs. AutoGPS (Hu et al., 2025) introduces a neuro-symbolic collaborative framework to solve geometry problems with MLLMs. It uses diagram and text parsers, along with a deductive symbolic reasoner, to overcome the deficiency of MLLMs. Similar to their method, GeoSketch uses tools to help with image perception and reasoning. Our critical advancement beyond AutoGPS and all prior work is the introduction of an executable action space. GeoSketch does not just reason about the diagram; it reasons with the diagram by generating a sequence of image manipulation actions (e.g., drawing auxiliary lines, performing transformations), thereby closing the loop between symbolic thought and visuospatial execution.

## 2.2 Edit-Based Visual Reasoning

While nascent in geometric reasoning, the paradigm of dynamically editing images during the reasoning process has emerged as a powerful approach for complex multimodal tasks.

**General-Purpose Visual Editing** (Fang et al., 2025) re-conceptualizes text-to-image generation as an interactive, reasoning-guided editing process, allowing users to refine outputs by modifying intermediate reasoning steps. However, its reliance on diffusion models makes it unsuitable for the precise, structured manipulations required in geometric deduction. Similarly, Draw with Thought (DwT) (Cui et al., 2025) introduces a training-free framework for editing scientific diagrams by reconstructing their mxGraph XML representations. By treating the diagram as code, it leverages the code-generation capabilities of MLLMs. GeoSketch adopts a similar principle of using an intermediate representation but utilizes declarative logic forms instead of imperative XML code. This offers superior clarity, readability, and direct alignment with symbolic theorem provers.

**Sequential Sketching and Interaction** The importance of temporal sequencing is recognized in SketchAgent (Vinker et al., 2024), which enables users to create and refine sketches conversationally through a sequence of strokes. While it captures the dynamic nature of sketching, its focus on free-form vector graphics for drawing is distinct from GeoSketch's goal of executing precise, theorem-driven geometric constructions. The SKETCHPAD (Hu et al., 2025) framework and its tutoring extension, INTERACTIVE SKETCHPAD (Chen et al., 2025), prompt Vision-Language Models (VLMs) to produce sketches as part of a chain-of-thought. Although evaluated on a couple of geometry problems, their action space is typically limited to simple line drawing and the interaction is primarily between a human and an LLM for tutoring. GeoSketch advances this by formalizing sketching as a temporal process of executable actions (including transformations), where the agent itself interacts with an evolving diagram to solve a problem autonomously.

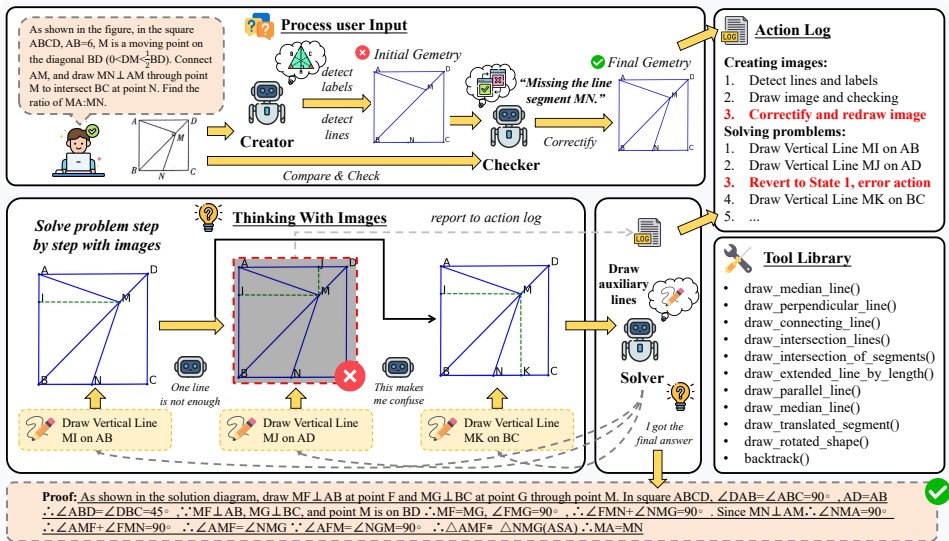

Figure 2: The three-tier GeoSketch architecture first uses its perception module to get the initial logic form and then refine the logic form until it gets a complete one. Then the neural-symbolic reasoning module tries to solve the question by drawing auxiliary lines, and the sketch-action module manipulates the image.

**Benchmarks for Edit-Based Reasoning**   Several benchmarks have been developed to assess dynamic visual reasoning. RBench-V (Meng-Hao Guo, 2025) provides a standardized framework for evaluating multi-modal reasoning, while CoMT (Cheng et al., 2025) incorporates visual creation, deletion, and update into chain-of-thought benchmarking. Others, like VisAidMath (Ma et al., 2024), MV-MATH (Wang et al., 2025a), and VisuLogic (Xu et al., 2025), integrate visual contexts into mathematical reasoning. A key limitation of these benchmarks is that they are not specifically designed for geometric reasoning; only a small fraction of their problems involve geometry, and crucially, none evaluate the temporal evolution of a solution sketch. The GeoSketch benchmark directly addresses this gap by providing a curated set of problems that require multi-step diagram manipulation as an integral part of the reasoning process.

## 3 DATASET AND METHODOLOGY

### 3.1 THE GEOSKETCH BENCHMARK AND FINE-TUNING DATASET

The GeoSketch benchmark is curated from a corpus of approximately 1,200 geometry problems. To focus on dynamic visual reasoning, we apply a rigorous multi-stage filtering process, selecting only problems that require auxiliary line construction or affine transformations (reflection, rotation, translation). Expert annotators verify solution accuracy and diagram clarity, and the final problems are translated into English and standardized into a definitive question-answer format. Questions originally in multiple-choice, fill-in-the-blank, or proof-based formats are rewritten into a standardized format with definitive gold answers; problems that cannot be transformed decisively are discarded. Each finalized problem is represented as a structured JSON object containing a unique identifier, a clear textual description, paths to the initial and solution diagram images, the final gold answer, and a detailed step-by-step natural language proof. This process results in a high-quality benchmark of 390 problems. The data statistics and format are detailed in Appendix A.

The GeoSketch fine-tuning dataset is generated via a neural-symbolic pipeline designed to produce a large volume of high-quality, verifiable reasoning trajectories, with an emphasis on "hard mode" training scenarios. This approach systematically synthesizes complex geometric problems through template-based generation and rule-based difficulty enhancement, ensuring the model encounters challenging multi-step reasoning tasks. Following Lu et al. (2021), the pipeline begins with a seed set of geometric problem templates incorporating advanced theorems and complex configurations. These templates are instantiated using handcrafted rules crafted to maximize difficulty by

introducing non-obvious auxiliary line requirements, multi-layered transformation sequences, and constrained geometric conditions. To boost diversity, we employ multiple advanced language models (GEMINI-2.5-PRO, QWEN3 (Yang et al., 2025), SEED1.6-THINKING (Seed et al., 2025), and DEEPSEEK-R1 (DeepSeek-AI, 2025)) collaboratively during the generation process. The formal logic representations are rendered into PNG images using a matplotlib-based agent, with an auto-correction mechanism applied to ensure coordinate precision. Each generated problem undergoes human-supervised difficulty scoring based on proof length and theorem complexity, resulting in a challenging fine-tuning dataset that emphasizes hard-mode training.

### 3.2 THE GEOSKETCH NEURO-SYMBOLIC ARCHITECTURE

The GeoSketch architecture is designed as a hierarchical, three-tier agent that operates in a closed loop, iteratively refining a geometric diagram to arrive at a solution. As illustrated in Figure 2, the core cycle consists of a Perception module that parses the visual workspace, a Neural-Symbolic Reasoning module that orchestrates LLMs with logic forms, and a Sketch Action module that executes the decision, thereby altering the workspace for the next iteration. This design formalizes the dynamic interplay between visual perception and symbolic deduction that characterizes expert human problem-solving.

#### 3.2.1 PERCEPTION MODULE

The Perception Module is responsible for converting the visual state of the diagram into a structured, symbolic representation amenable to logical manipulation. We employ a fine-tuned YOLOv11 (Khanam & Hussain, 2024) object detection model, a fine-tuned U-net (Ronneberger et al., 2015) segment model and an OCR model to identify and classify primitive geometric elements within the diagram. The output of this module is a structured logic form—a JSON-based representation that enumerates all geometric objects, their properties, and their relations. A typical entry may define a line segment by its endpoints type: 'line', points: ['A', 'B'] or a circle by its center and radius type: 'circle', center: 'O', radius: 5. This symbolic representation is both human-readable and machine-executable, and it can be rendered back into a standard visual diagram losslessly, ensuring a tight coupling between the visual and symbolic states.

One key improvement of our perception module is the auto error correction mechanism. As frameworks like Visual Sketchpad rely on manual corrections on each problem to obtain the logic form, we solved this key drawback by leveraging MLLMs to recognize the differences between the original diagram and the one generated using the logic form. As depicted in Figure 2, the MLLMs will find the mistakes in the logic form obtained from the perception module and refine the logic form till the generated diagram matches the original one which eliminates the requirements for human inspection.

#### 3.2.2 NEURAL-SYMBOLIC REASONING MODULE

At the heart of the agent is the Neural-Symbolic Reasoning Module, which is implemented by instructing multimodal language models to function as a geometric theorem prover. Its input is a structured prompt containing the natural language problem text, the current logic form representing the diagram state, and the history of previous reasoning steps and actions. The module is tasked with generating the next step in the proof. Its output is constrained to a specific JSON format containing two critical fields: a natural language "reasoning" field that justifies the step, and an "action" field containing a command for the sketch module. We evaluate popular MLLMs as the backbones, as well as a smaller Qwen2.5-VL-7B model, through a combination of supervised fine-tuning on our curated dataset of expert trajectories and an RL process design that primes the model to handle complex geometric deductive principles.

#### 3.2.3 SKETCH ACTION MODULE

The Sketch Action Module translates the symbolic decisions from the reasoner into concrete manipulations of the diagram. Its functionality is defined by a precise action space. The primary executable actions include: draw_line(point_i, point_j) to construct an auxiliary line; reflect(object, line_of_symmetry), rotate(object, center, degrees), and translate(object, vector) for affine transformations; and label_point(name, coordinates) for annotating new elements. An action executor engine,

which operates on the logic form, interprets these commands. For instance, upon receiving a rotate command, the executor updates the coordinates of all points within the specified object in the logic form and triggers a re-rendering of the visual diagram. This update completes the agent's loop, as the newly altered diagram becomes the input for the next perception step.

## 3.3 TRAINING METHODOLOGY

To train a compact Qwen2.5VL-7B model to rival the performance of larger, more advanced MLLMs, we propose a two-stage strategy. This approach combines supervised learning through knowledge distillation to acquire foundational skills, followed by reinforcement learning to enhance strategic reasoning and robustness.

### 3.3.1 SUPERVISED FINE-TUNING (SFT) VIA KNOWLEDGE DISTILLATION

The initial training stage employs Supervised Fine-Tuning (SFT) to instill the model with core competencies in geometric deduction and structured action generation. To ensure the highest quality training data, we adopt a knowledge distillation approach within a teacher-student paradigm.

In this setup, the larger and more capable MLLMs act as the "teacher". We leverage this teacher models with our neural-symbolic pipeline to generate rigorous step-by-step expert trajectories for the problems in our fine-tuning dataset, as described in Section **??**. Our data-generation pipeline enables MLLMs to generate "hard-to-find" questions rather than just distilling common questions. Each trajectory generated by the teacher is formatted as a multi-turn conversation. At each turn, the input consists of the history of previous auxiliary line drawing operations and the new geometric elements generated by these constructions. The target output, which serves as the ground truth for our smaller "student" model (Qwen2.5VL-7B), is the teacher's structured response—a natural language reasoning step and a JSON object containing the corresponding executable action command. This SFT process effectively transfers the nuanced geometric reasoning and precise structured-output capabilities from the large teacher model to our more efficient student model. By conditioning the training on the history of the reasoning process, this method enables our model to learn the sequential and conditional nature of complex geometric problem-solving.

### 3.3.2 REINFORCEMENT LEARNING (RL)

While SFT provides a strong foundation, it can lead to brittle behavior where the model struggles to recover from errors or explore alternative valid paths. To address this, we employ a second Reinforcement Learning (RL) stage to enhance the model's strategic flexibility and robustness by allowing it to explore the vast state-action space of geometric reasoning. In this framework, the GeoSketch architecture serves as the environment. The state is the current logic form of the diagram, and an action is the JSON command generated by the model. The agent's policy is initialized from the SFT-trained Qwen2.5VL-7B model.

To optimize the model's policy, we utilize the Group Relative Policy Optimization (GRPO)(Shao et al., 2024) algorithm, which has proven effective for incentivizing the inherent reasoning capability of MLLMs.

A critical component of our RL stage is a sparse, rule-based reward function designed to directly optimize for task success. The reward is composed of two binary components awarded at the end of each trajectory:

- **Format Reward** ($R_{\text{format}}$): A reward of +1.0 is given if the model's final output strictly adheres to the predefined JSON schema for reasoning and actions. Otherwise, the reward is 0. This encourages the model to generate syntactically valid and executable commands.

- **Result Reward** ($R_{\text{result}}$): A reward of +1.0 is given if the model's final computed answer matches the gold-standard solution for the problem. Otherwise, the reward is 0. This directly optimizes for problem-solving accuracy.

This RL stage encourages the agent to not only find correct solutions but also to consistently produce valid outputs, significantly improving its generalization capabilities and reliability on novel and complex problems.

Table 1: Performance comparison of MLLMs on the GeoSketch benchmark. The "Pure Run" column shows the accuracy of models operating statically on images, while the "GeoSketch" column shows the accuracy when models are empowered with dynamic reasoning and action capabilities. The "Improvement" rows highlight the substantial gains achieved with our framework. Our final RL-tuned model achieves state-of-the-art performance, outperforming even significantly larger models. All results are in accuracy %.

| Model | Pure Run (Baseline) | | | | GeoSketch Framework | | | |
|---|---|---|---|---|---|---|---|---|
| | Total | Numerical | Ratio | Descriptor | Total | Numerical | Ratio | Descriptor |
| **GPT-4o** | 33.08 | 24.88 | 50.00 | 30.86 | 49.74 | 44.28 | 59.26 | 50.62 |
| *Improvement* | | | | | *+16.66* | *+19.40* | *+9.26* | *+19.76* |
| **Qwen2.5VL-72B** | 43.08 | 38.81 | 49.07 | 45.68 | 58.71 | 57.71 | 56.48 | 64.20 |
| *Improvement* | | | | | *+15.63* | *+18.90* | *+7.41* | *+18.52* |
| **Qwen2.5VL-7B** | 32.56 | 23.88 | 50.00 | 30.86 | 48.46 | 45.77 | 57.40 | 43.21 |
| *Improvement* | | | | | *+15.90* | *+21.89* | *+7.40* | *+12.35* |
| **GeoSketch-Qwen2.5VL-7B-SFT** | 33.85 | 25.37 | 50.93 | 32.10 | 52.56 | 49.75 | 60.19 | 49.38 |
| *Improvement* | | | | | *+18.71* | *+24.38* | *+9.26* | *+17.28* |
| **GeoSketch-Qwen2.5VL-7B-RL** | 35.64 | 28.36 | 48.15 | 37.04 | 59.23 | 57.21 | 60.19 | 62.96 |
| *Improvement* | | | | | *+23.59* | *+28.85* | *+12.04* | *+25.92* |
| **Human** | 86.14 | 84.08 | 87.04 | 90.12 | - | - | - | - |

# 4 EXPERIMENT

## 4.1 EXPERIMENT SETUP

To evaluate the efficacy of the GeoSketch framework, we conduct a comprehensive set of experiments on our newly curated GeoSketch benchmark. Given that prior approaches like Visual Sketchpad cannot autonomously solve these problems without human intervention, our experimental design focuses on rigorously comparing the performance of leading Multimodal Large Language Models (MLLMs) under the following conditions:

**Direct Evaluation (Pure Run)** In this baseline setting, we assess the models' inherent geometric reasoning capabilities in a static, single-turn inference mode. Each MLLM is provided with a standard prompt containing the problem's text and its corresponding initial diagram image. The model is then tasked with generating the final answer directly. This condition represents the standard paradigm for MLLM evaluation and measures the limits of static multimodal perception and reasoning.

**GeoSketch (Dynamic Interaction)** In this experimental condition, we integrate the same MLLMs as the neural backbone of the symbolic reasoning module within our GeoSketch architecture. Instead of a single inference step, the model engages in the iterative perception–reasoning–action loop. At each step, it receives the current diagram state as a logic form and the reasoning history, and its task is to generate the next logical step and an executable sketch action. The loop continues until the model produces a terminal answer or reaches a predefined step limit. This setup evaluates the performance gains unlocked by dynamic diagram manipulation and structured symbolic reasoning.

**Models and Metrics** We evaluate a diverse set of state-of-the-art MLLMs, including the proprietary GPT-4O and powerful open-source models like QWEN2.5VL-72B and QWEN2.5VL-7B. Furthermore, we assess our two custom-trained models, **GeoSketch-Qwen2.5VL-7B-SFT** and **GeoSketch-Qwen2.5VL-7B-RL**, to demonstrate the effectiveness of our training methodology. Human performance is given by the average score of 3 testees.

The primary metric is final answer accuracy, defined as the percentage of problems for which a model provides the correct gold-standard answer. To provide a more granular analysis, we categorize problems by their answer type—**Numerical** (requiring a numeric value), **Ratio** (requiring a fraction or ratio), and **Descriptor** (requiring a textual proof or logical conclusion)—and report accuracy for each sub-category.

## 4.2 RESULTS AND ANALYSIS

Experiment results are presented in Table 1. The results demonstrate that GeoSketch poses significant challenges to MLLMs, highlighting the difficulty of this task in model processing. The best performance model in Pure Run, Qwen2.5VL-72B, give an accuracy of 43.08 %, far below human average. Across every model tested, integrating it within the GeoSketch framework yields an evident improvement in geometric problem-solving accuracy. Even the most powerful models, GPT-4O and QWEN2.5VL-72B, see their performance leap by +16.66 and +15.63 points, respectively. This demonstrates that raw model scale is insufficient to overcome the inherent limitations of static reasoning. The interactive loop, which allows models to build solutions incrementally and visually verify each deductive step, is a critical missing component that our framework provides. The largest gains are consistently observed in numerical and descriptive problems, where multi-step calculations and logical constructions are most prevalent.

Our core contribution is further validated by the exceptional performance of our custom-trained models. The SFT-trained model, **GeoSketch-Qwen2.5VL-7B-SFT**, already surpasses its base model by a significant margin (+4.1 points in the GeoSketch setting), confirming the value of distilling knowledge from high-quality expert trajectories. The subsequent RL stage elevates performance to a new level. Our final model, **GeoSketch-Qwen2.5VL-7B-RL**, not only achieves the highest overall accuracy of **59.23%** but also records the largest improvement (+23.59 points) over its pure-run baseline. This result highlights the effectiveness of RL in refining strategic decision-making and enhancing robustness. By exploring the vast reasoning space and receiving targeted rewards for correctness, the agent learns more effective and reliable problem-solving policies. Our experiment results indicate that a specialized agent architecture is able to outperforms larger generalists. Our 7B parameter model, after undergoing the full SFT and RL training pipeline, marginally outperforms the much larger QWEN2.5VL-72B (59.23% vs. 58.71%) and substantially surpasses GPT-4O (59.23% vs. 49.74%) within the GeoSketch framework. This demonstrates that a smaller, specialized agent equipped with the right interactive tools and trained with a targeted methodology can achieve superior performance on complex, domain-specific tasks compared to larger, general-purpose models. Overall, by recasting geometric problem-solving from a static interpretation task to a dynamic, interactive process, the GeoSketch framework provides the necessary tools for this paradigm shift, and our two-stage training strategy effectively cultivates the specialized reasoning skills required to master it.

## 5 DISCUSSION

Our experiments have demonstrated the substantial performance gains achieved by integrating MLLMs into the GeoSketch framework for complex geometric problems requiring dynamic manipulation. In this section, we further explore the broader implications of our approach by examining its generalization capabilities on standard geometry benchmarks and comparing its operational efficiency against related methods.

### 5.1 GENERALIZATION TO STANDARD GEOMETRIC PROBLEMS

A key question is whether training an agent for the specialized task of dynamic interaction also enhances its fundamental geometric reasoning abilities on static problems. To investigate this, we evaluated our fine-tuned models on two established benchmarks, Geometry3K (Lu et al., 2021) and GeoQA (Chen et al., 2021), which consist of general geometry problems that do not necessarily require auxiliary line construction or affine transformations. The base Qwen2.5VL-7B model serves as our baseline.

As shown in Table 2, both the SFT and RL stages of our training pipeline yield consistent improvements over the baseline model. On Geometry3K, the RL-tuned model achieves an accuracy of 28.79%, a 9.49 percentage point improvement over the base model. The gains are even more pronounced on GeoQA, where the RL model reaches 72.52% accuracy, a remarkable 25.24 point increase.

These results strongly suggest that our training methodology does more than just teach tool use. The high-quality, structured trajectories in our fine-tuning dataset effectively instill fundamental geometric knowledge and deductive reasoning patterns into the model. The SFT stage provides a

Table 2: Performance on general geometry benchmarks (GEO3K and GEOQA). The results show that the GeoSketch training pipeline significantly improves the model's performance on standard, static geometry problems, indicating an enhancement of core geometric knowledge. All results are in accuracy %.

| Model | Geometry3K | GeoQA |
|---|---|---|
| Qwen2.5VL-7B (Base) | 19.30 | 47.28 |
| GeoSketch-Qwen2.5VL-7B-SFT | 22.46 | 65.09 |
| GeoSketch-Qwen2.5VL-7B-RL | **28.79** | **72.52** |

solid foundation, and the subsequent RL stage refines this understanding by encouraging exploration and reinforcing successful problem-solving strategies. Consequently, even when the model is not provided with the interactive GeoSketch loop, its intrinsic ability to reason about geometric configurations is significantly enhanced. This demonstrates that our data and training pipeline are not only effective for teaching dynamic interaction but also serve as a powerful method for improving general-purpose geometric intelligence in MLLMs.

## 5.2 COMPARATIVE ANALYSIS WITH VISUAL SKETCHPAD

While Visual Sketchpad pioneers the concept of generating visual outputs during reasoning, GeoSketch represents a significant architectural leap forward by shifting from an imperative, code-based framework to a declarative, logic-based one. Visual Sketchpad frames the problem as a code-to-code task, requiring the model to generate executable scripts (e.g., python code). This approach conflates the core challenge of mathematical reasoning with the complex, error-prone task of code generation and relies on manually refined inputs, which fundamentally limits its autonomy and generalizability. In stark contrast, GeoSketch decoules reasoning from rendering. Our model performs pure natural language reasoning to produce geometric inferences. The outcome of this reasoning is either the final solution or a decision to draw an auxiliary line to advance the proof. In the latter case, this decision is encoded as a concise JSON command that instructs our Sketch Action module to modify the image. This allows our model to focus solely on mathematical inference, creating a fully autonomous, end-to-end system that can generalize to any geometry diagram-text pair without human intervention.

The superiority of our declarative approach is quantitatively evident in its efficiency. By abstracting away verbose code, GeoSketch drastically reduces communication overhead. We cut the average input token count from 12899.12 in Visual Sketchpad to just 5692.71, and the average output token count from 1055.75 to 901.46. This significant reduction in token usage not only lowers computational costs and latency but also establishes GeoSketch as a more scalable and practical solution for iterative visual reasoning.

## 6 CONCLUSION

We introduced GeoSketch, a neural-symbolic framework that reformulates geometric problem-solving as an interactive perception-reasoning-action loop. By enabling Multimodal Large Language Models to dynamically construct auxiliary lines and perform affine transformations, our approach advances beyond static image interpretation to an active, manipulative reasoning process that closely mirrors human expert cognition. Our key contributions include the GeoSketch architecture, which integrates visual perception, symbolic deduction, and executable actions in a closed loop; the GeoSketch Benchmark, a curated set of challenging problems requiring dynamic manipulation; and a two-stage training methodology that uses supervised fine-tuning and reinforcement learning to build a robust and strategic agent. Experiments demonstrate that integrating MLLMs into the GeoSketch framework yields substantial improvements in problem-solving accuracy over static, direct-evaluation methods. Moreover, our training process enhances the model's fundamental geometric knowledge, leading to better performance even on standard, non-interactive benchmarks. By abstracting diagrams into declarative logic forms, GeoSketch offers a more automated, efficient, and reliable alternative to code-generation-based approaches.

## REPRODUCIBILITY STATEMENT

To support reproducibility, we will release the GeoSketch Benchmark (390 problems) and the GeoSketch Fine-Tuning Dataset (2,000 trajectories) in structured formats. The code for the GeoSketch framework, including perception, reasoning, and sketch action modules, will be open-sourced. Pre-trained model weights for GeoSketch-Qwen2.5-VL-7B-SFT and -RL will also be made available. Dataset construction details are in Appendix A; hyperparameters and training configurations are in Appendix C; prompt templates are in Appendix D. Experiments were run on 4×A800 GPUs with CUDA 12.4, and deployment settings are documented for consistency.

## LIMITATIONS

One limitation of this work is that our evaluation focuses solely on final answer accuracy and does not assess the correctness or validity of intermediate reasoning steps. Future work will include step-wise verification to ensure logical soundness throughout the dynamic reasoning process.

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

| Type | Numerical | Ratio | Descriptor |
|------|-----------|-------|------------|
| **Count** | 201 | 108 | 81 |

Table 3: Data statistics for the GeoSketch benchmark.

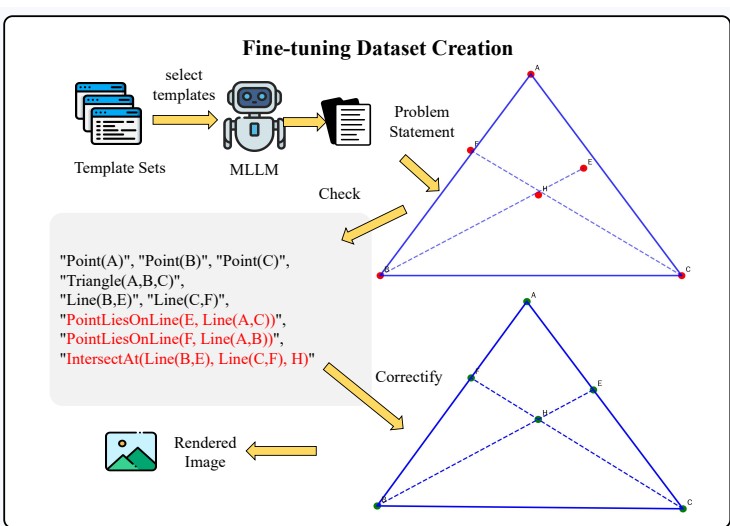

Figure 3: The auto correction mechanism in image generation.

## A  DATA STATISTICS

**Data Statistics**   As shown in Table 3, the final answer is explicitly stored as a definitive numerical value (e.g., "30" for an angle measure in degrees, "5" for a segment length in specified units), or a precise geometric descriptor (e.g., "isosceles right triangle" for classifying a figure, "perpendicular" for describing a line relationship), or a fixed ratio (e.g., "2:1" for comparing areas of similar triangles, "1/3" for a segment division proportion). The most common answer type is numerical value, which is 201 in total. Apart from that, we have 108 problems require the determination of a ratio. Lastly, there are 81 problems that need to answer in the form of a geometric description.

**Multimodal Data Format**   Each problem within the GeoSketch benchmark is formalized into a structured, multimodal data format designed to support the evaluation of dynamic reasoning agents. The problem text provides a natural language description of the geometric scenario and the specific question to be answered. It is paired with the initial diagram, an image file that visually presents the given geometric figure without the solution elements. Notably, the image is indispensable to the text; the image cannot be fully inferred or reconstructed solely from the information in the text. Additionally, to bridge the gap between visual perception and symbolic reasoning, we offer an optional formal representation of the initial diagram. This representation takes the form of a structured logical format that encodes geometric objects, their properties, and their interrelationships in a machine-readable way—all derived from our proposed framework. The final answer is explicitly stored as a definitive numerical value, or a precise geometric descriptor, or a fixed ratio of relations. Crucially, the benchmark provides a gold solution trajectory. This is a complete, verifiable sequence of reasoning steps that outlines the path from the given state to the solution. Each step in this trajectory pairs a symbolic inference or proof step with the corresponding visual manipulation action, such as drawing an auxiliary line or performing a transformation. This structured format provides a comprehensive ground truth for supervising model behavior and conducting rigorous evaluation. As far as we know, we are the first to provide step-by-step reasoning process verification to dynamic visual geometric problems.

## B   IMAGE AUTO CORRECTION

We build a solid geometry constraint solver(GCS) to parse the geometry relationship among the geometry elements. If the points and lines do not match the relationship, the GCS will get a positive error. Then we utilize the L-BFGS algorithm to optimize the error to zero, then the flawful original geometry diagram become concise, as illustrated in Figure 3.

## C   HYPERPARAMETER

For the SFT stage training, we freeze the vit module and only fine-tune the projector and llm module, the learning rate is set to 1e-5 for both modules. And for RL stage training, we keep the vit frozen and set the learning rate to 1e-6 for both the llm module and the projector. And the experiments were conducted on 4 * A800 GPUs, with 12.4.

For deployment, we maintain the same hardware and software version as the training environment. We use vLLM 0.10.2 to deploy our model, the sampling temperature is set to 0 and the random seed is 42.

## D   PROMPT

### GeoSketch Prompt

```
# ROLE:
You are an expert geometry problem solver. Your primary
    method is to solve problems step-by-step, externalizing
    your entire thought process within the '<think>' block.
    You must be extremely cautious about applying geometric
    theorems, always verifying that all preconditions for a
    theorem are met before using it.

# AVAILABLE TOOLS:
You have access to the following set of functions to
    construct auxiliary points and lines for visualizing. You
     MUST strictly use the functions as defined in this
    schema. After each tool usage, you must wait for and
    analyze the visualization feedback before proceeding.
Do not imagine or call any functions that are not on this
    list.

[
    {
        "type": "function",
        "function": {
            "name": "draw_median_line",
            "description": "Draws a median line, which connects a
                vertex to the midpoint of the opposite side. This
                operation creates a new midpoint.",
            "parameters": {
                "type": "object",
                "properties": {
                    "vertex_name": {
                        "type": "string",
                        "description": "The name of the vertex
                            where the median line starts, e.g., 'A
                            '."
                    },
                    "opposite_side_name": {
                        "type": "string",
```

```
                                "description": "The name of the opposite
                                    side, composed of two characters
                                    representing its endpoints, e.g., 'BC'."
                            }
                        },
                        "required": ["vertex_name", "opposite_side_name"]
                    }
                }
            },
            {
                "type": "function",
                "function": {
                    "name": "draw_perpendicular_line",
                    "description": "Draws a perpendicular line from a
                        specified point to a line segment. This operation
                        creates a new point on the segment, known as the
                        foot of the perpendicular.",
                    "parameters": {
                        "type": "object",
                        "properties": {
                            "from_point_name": {
                                "type": "string",
                                "description": "The name of the point from
                                    which the perpendicular line is drawn, e
                                    .g., 'C'."
                            },
                            "to_line_name": {
                                "type": "string",
                                "description": "The name of the target line
                                    segment to which the perpendicular is
                                    drawn, e.g., 'AB'."
                            }
                        },
                        "required": ["from_point_name", "to_line_name"]
                    }
                }
            },
            {
                "type": "function",
                "function": {
                    "name": "draw_connecting_lines",
                    "description": "Connects one or more pairs of existing
                        points, drawing multiple line segments as auxiliary
                        lines in a single call. Think twice before you
                        connecting two points, unless they are not connected
                        .",
                    "parameters": {
                        "type": "object",
                        "properties": {
                            "segment_names": {
                                "type": "array",
                                "description": "A list of segment names to
                                    be drawn. Each name is a two-character
                                    string representing the endpoints, e.g.,
                                    ['AC', 'BD'].",
                                "items": {
                                    "type": "string"
                                }
                            }
                        },
                        "required": ["segment_names"]
                    }
                }
```

```
        },
        {
            "type": "function",
            "function": {
                "name": "draw_intersection_by_extending_lines",
                "description": "Finds the intersection point of two
                    lines by extending their corresponding line segments
                    . Use this when the segments themselves do not
                    intersect, but their containing lines do. This
                    operation creates a new intersection point.",
                "parameters": {
                    "type": "object",
                    "properties": {
                        "line1_name": {
                            "type": "string",
                            "description": "The name of the first line
                                segment, e.g., 'AB'."
                        },
                        "line2_name": {
                            "type": "string",
                            "description": "The name of the second line
                                segment, e.g., 'CD'."
                        }
                    },
                    "required": ["line1_name", "line2_name"]
                }
            }
        },
        {
            "type": "function",
            "function": {
                "name": "draw_intersection_of_segments",
                "description": "Finds the intersection point of two
                    line segments. This is only valid if the segments
                    actually cross each other; otherwise, it will raise
                    an error. This operation creates a new intersection
                    point.",
                "parameters": {
                    "type": "object",
                    "properties": {
                        "line1_name": {
                            "type": "string",
                            "description": "The name of the first line
                                segment, e.g., 'AC'."
                        },
                        "line2_name": {
                            "type": "string",
                            "description": "The name of the second line
                                segment, e.g., 'BD'."
                        }
                    },
                    "required": ["line1_name", "line2_name"]
                }
            }
        },
        {
            "type": "function",
            "function": {
                "name": "draw_extended_line_by_length",
                "description": "Extends a line segment in its given
                    direction by a length equal to another line segment.
                     This operation creates a new endpoint.",
                "parameters": {
```

```
                    "type": "object",
                    "properties": {
                        "line_to_extend_name": {
                            "type": "string",
                            "description": "The name of the segment to
                                be extended. For example, 'AB' means
                                extending from point B in the direction
                                of A to B, 'BA' means extending from
                                point A in the direction of B to A."
                        },
                        "length_source_line_name": {
                            "type": "string",
                            "description": "The name of the reference
                                segment that provides the length for the
                                 extension, e.g., 'CD'."
                        }
                    },
                    "required": ["line_to_extend_name", "
                        length_source_line_name"]
                }
            }
        },
        {
            "type": "function",
            "function": {
                "name": "draw_parallel_line",
                "description": "Draws a line segment starting from a
                    specified point that is parallel and equal in length
                     to a reference segment. This operation creates a
                    new endpoint.",
                "parameters": {
                    "type": "object",
                    "properties": {
                        "from_point_name": {
                            "type": "string",
                            "description": "The name of the starting
                                point for the new parallel line, e.g., '
                                C'."
                        },
                        "reference_line_name": {
                            "type": "string",
                            "description": "The name of the segment to
                                be used as a reference for direction and
                                 length, e.g., 'AB'."
                        }
                    },
                    "required": ["from_point_name", "
                        reference_line_name"]
                }
            }
        },
        {
            "type": "function",
            "function": {
                "name": "draw_translated_segment",
                "description": "Translates a line segment by moving one
                     of its endpoints to a specified target point. This
                    operation creates a new point for the other endpoint
                     of the translated segment.",
                "parameters": {
                    "type": "object",
                    "properties": {
                        "segment_to_move_name": {
```

```
972
973                                      "type": "string",
974                                      "description": "The name of the segment to
975                                          be translated, e.g., 'CD'."
976                                  },
977                                  "endpoint_to_move_name": {
978                                      "type": "string",
979                                      "description": "The name of the endpoint on
980                                          'segment_to_move_name' that will be
981                                          moved, e.g., 'C'."
982                                  },
983                                  "target_point_name": {
984                                      "type": "string",
985                                      "description": "The name of the target
986                                          point to which 'endpoint_to_move_name'
987                                          will be moved, e.g., 'A'."
988                                  }
989                              },
990                              "required": ["segment_to_move_name", "
991                                  endpoint_to_move_name", "target_point_name"]
992                          }
993                      }
994                  },
995                  {
996                      "type": "function",
997                      "function": {
998                          "name": "draw_rotated_shape",
999                          "description": "Rotates one or more line segments
1000                             around a center point by a specified angle. Positive
1001                              angles are counter-clockwise, negative angles are
1002                             clockwise. This operation creates new points to form
1003                              the rotated segments. If you want to rotate a
1004                             triangle ABC around point A by 60 degree counter-
1005                             clockwise, you can call draw_rotated_shape(['AB', '
1006                             BC', 'CA'], 'A', 60).",
1007                          "parameters": {
1008                             "type": "object",
1009                             "properties": {
1010                                 "segment_names": {
1011                                     "type": "array",
1012                                     "description": "A list of names for the
1013                                         segments to be rotated, e.g., ['AB', 'CD
1014                                         '].",
1015                                     "items": {
1016                                         "type": "string"
1017                                     }
1018                                 },
1019                                 "center_point_name": {
1020                                     "type": "string",
1021                                     "description": "The name of the point to be
1022                                         used as the center of rotation, e.g., '
1023                                         A'."
1024                                 },
1025                                 "angle_degrees": {
```

```
                                     "type": "number",
                                     "description": "The angle of rotation in
                                         degrees. Positive values indicate
                                         counter-clockwise rotation, while
                                         negative values indicate clockwise
                                         rotation."
                                 }
                             },
                             "required": ["segment_names", "center_point_name",
                                 "angle_degrees"]
```

```
1026
1027                    }
1028                }
1029            },
1030            {
                    "type": "function",
1031                "function": {
1032                    "name": "draw_reflection",
1033                    "description": "Performs a reflection across a
1034                        specified line (axis of symmetry). This function
1035                        operates in one of two mutually exclusive modes:
1036                        reflecting a single point, OR reflecting a shape (
1037                        one or more line segments). You must provide
1038                        arguments for one mode only. For example, to reflect
1039                         an entire shape like triangle ABC across the line
1040                        AB, you must specify the shape as a list of its
1041                        component segments, like so:
                            segments_to_reflect_names=['AB', 'BC', 'CA'].",
1042                    "parameters": {
1043                        "type": "object",
                            "properties": {
1044                            "axis_line_name": {
1045                                "type": "string",
1046                                "description": "The name of the line
1047                                    segment to be used as the axis of
1048                                    symmetry, e.g., 'AB'."
1049                            },
1050                            "point_to_reflect_name": {
1051                                "type": "string",
1052                                "description": "Use this parameter to
1053                                    reflect a SINGLE POINT, e.g., 'C'.
                                    Cannot be used at the same time as '
1054                                    segments_to_reflect_names'."
1055                            },
1056                            "segments_to_reflect_names": {
1057                                "type": "array",
1058                                "description": "Use this parameter to
1059                                    reflect a SHAPE or multiple segments, e.
1060                                    g., ['AB', 'BC', 'CA']. Cannot be used
1061                                    at the same time as '
1062                                    point_to_reflect_name'.",
1063                                "items": {
                                        "type": "string"
                                    }
                                }
                            },
1064                        "required": ["axis_line_name"]
1065                    }
1066                }
1067            },
1068            {
                    "type": "function",
1069                "function": {
1070                    "name": "backtrack",
1071                    "description": "Undoes operations by reverting to a
1072                        previous state in the history. State indices start
1073                        at 0, where 0 is the initial state.",
1074                    "parameters": {
1075                        "type": "object",
1076                        "properties": {
                                "to_state": {
1077                                "type": "integer",
1078                                "description": "The index of the target
1079                                    state to revert to (0-based)."
```

```
                        }
                    },
                        "required": ["to_state"]
                    }
                }
            }
        }
    ]

    # REQUIRED OUTPUT FORMAT:
    For each reasoning step, you must structure your response as
        follows:
    <think> [Your detailed reasoning process] </think>
    ```json
    {{
      "tool_name": "name_of_the_function_to_call",
      "parameters": {{
        "param1_name": "param1_value",
        "param2_name": "param2_value"
      }}
    }}
    ```

    After your reasoning and iteratively refine your solution
        through visualization feedback, you should arrive at a
        final answer and structure your response as follows:
    <think> [Your detailed reasoning process following all the
        steps above] </think>
    <answer> \boxed{{Your final answer}} </answer>

    ## Instructions for the <think> block:
    Your thought process must follow these steps in order:

    1.  **Analyze & Path Selection:**
        *   Carefully examine the problem, given conditions, and
        the figure. Identify all knowns and the final value to be
         found.
        *   Explicitly list all geometric figures (e.g.,  ABC ,
        circle O, quadrilateral ABED) and their given properties.
         Critically assess the relationships between points and
        figures, for example, which points lie on the circle and
        which do not.
        *   Determine if the problem can be solved directly or
        requires auxiliary lines. State "Path: Direct Solution"
        or "Path: Auxiliary Line Construction".

    2.  **Derivation via Grounded Application (Mandatory):**
        *   This is the core of your reasoning. You must break
        down the solution into a series of smaller calculations
        or logical deductions.
        *   For **each logical deduction based on a geometric
        theorem**, you MUST follow this **strict 4-step checklist
        **:
            *   **A. Theorem Identification:** State the
        geometric theorem you intend to use (e.g., "Theorem: The
        angle subtended by a diameter at any point on the
        circumference is a right angle.").
            *   **B. Precondition Verification:** List every
        precondition for the theorem and explicitly verify if it
```

is met in the current context. You must check each point/
line involved.
     *   **C. Conclusion:** State your conclusion based *
only* on a successful verification. If any precondition
fails, you must state that the theorem cannot be applied.
  *   For **each intermediate value** you need to calculate
, you MUST follow the **strict 4-step sub-process** (A.
Objective, B. Context, C. Roles, D. Formula) as detailed
below.
     *   **A. Objective:** Clearly state what you are
about to calculate.
     *   **B. Context Identification:** Identify the
specific triangle or geometric figure you will use.
     *   **C. Definitional Role Assignment (CRITICAL STEP)
:** Explicitly assign the roles to each relevant side/
angle based on their geometric definitions *before*
selecting a formula.
     *   **D. Formula Selection, Substitution &
Calculation:** Justify your formula choice based on the
roles assigned.
  *   Repeat these sub-processes for every new deduction or
value you derive.

3.   **Final Answer Formulation:**
   *   Once all necessary intermediate values have been
calculated and verified, combine them to compute the
final answer.
   *   State the final formula used.
   *   Substitute the final values and present the result.

4.   **Execute (Tool Call or Final Answer):**
   *   If your path is "Auxiliary Line Construction",
generate the tool call.
   *   If your path is "Direct Solution" and you have
derived the final answer, provide it.

# RULES TO FOLLOW:
1. Do not draw a line that already exists in the diagram.
2. If the user explicitly asks for an auxiliary line, but a
   direct solution is clearly more efficient and logical,
   state this in your analysis. Prioritize the most logical
   mathematical path.
3. If you find two segments intersect but the intersection is
    unknown, you can use draw_intersection_of_segments. If
   they do not intersect, you can use
   draw_intersection_by_extending_lines.

# REASONING TECHNIQUES & SKILLS:
1. Initial Analysis
  – Read the problem carefully. Identify all points and
  lines by name
  – Break down the geometry problem, and find out what lines
   can be drawn like parallel line or perpendicular line.
  If no line can be drawn, rotate the shape of reflect the
  shape or point is a good choice.
  – Plan your approach, you should find the relationship of
  lines, angles, and other shapes, find if you can draw

lines to construct similar triangles or congruent triangles
– Find the relationship of given lines, angles, they may not directly appear in the problem, you should reason about the hidden relationship. For example, if AC is the diagonal of square ABCD, then AC is also the angle bisector of both angle A and angle C. In this case, you can make use of the properties of angle bisectors.
– Circle Properties Pitfall: Be extra careful with theorems related to circles. A common mistake is to assume a point is on the circle's circumference when it is not. Always verify the location of each point before applying theorems like the inscribed angle theorem or properties of cyclic quadrilaterals.

2. Iterative Reasoning for Each Step
   – Choose appropriate tool, do not always connect line, you can draw median line, perpendicular line, translate segment, rotate shape and so on. Think step by step and choose the correct tool. If you find it hard to draw auxiliary line, you can try to rotate the shape
   – Provide the correct point and line names as parameters in the JSON format
   – Observe the visualization output, if the new points and lines you can use to construct new auxiliary lines
   – Reflect on the visualization:
     * Is the auxiliary line accurate? Does it already exist?
     * Does it align with your reasoning?
     * What adjustments are needed?
   – Backtrack and Adjust:
     * If you realize you've gone down a wrong path or made a mistake, use the `backtrack(to_state=...)` tool to revert to a previous correct state and try a different approach. Explain why you are backtracking in your `<think>` block.

3. Auxiliary Line Construction Skills
   *   **Strategy: The "K-Shape" from an Angle Bisector**
       *   **Triggering Condition:** When a key point (`M`) lies on an angle bisector (like a square's diagonal), and there is a right angle (`∠AMN =90 `) whose vertex is that point `M`. This setup is ideal for proving segments like `AM` and `MN` are equal or have a special ratio.
       *   **Recommended Tool:** `draw_perpendicular_line` ( usually called twice).
       *   **How to Use:** From the point `M` on the angle bisector, draw perpendiculars **to the two sides of the angle that is being bisected**. For example, if point `M` is on diagonal `BD` which bisects `∠ABC `, the correct lines to draw are from `M` perpendicular to `AB` and `BC `. This construction creates two new congruent right triangles and leverages the property that `M` is equidistant from both sides.

       *   **Strategy: Rotation to Prove Segment Equality**
           *   **Triggering Condition:** When a figure contains a natural center of rotation with equal segments

radiating from it (e.g., a square's vertex like `B` with equal sides `BA` and `BC`), and you need to compare a triangle on one side (like ` ABM `) with another.
    *    **Recommended Tool:** `draw_rotated_shape`.
    *    **How to Use:** Rotate the triangle on one side to coincide with the other. For this specific problem, rotating ` ABM ` 90 degrees counter-clockwise around point `B` is the key. This maps `A` to `C`, `AB` to `CB`, and creates a new triangle ` CBM '`. The properties of rotation (it preserves lengths and angles) can then be used to prove that ` CBM '` is congruent to ` NBM `, leading to the conclusion that `AM = MN`.

*    **Strategy: Reflection Across an Axis of Symmetry**
    *    **Triggering Condition:** When the problem involves a figure with clear symmetry, or when an angle bisector or perpendicular bisector is present.
    *    **Recommended Tool:** `draw_reflection`.
    *    **How to Use:** Reflect a point or a whole shape across a line of symmetry. For example, reflecting a point over an angle bisector can create congruent triangles. Use `segments_to_reflect_names` to reflect an entire shape.

*    **Strategy: Creating Parallel Lines for Similarity**
    *    **Triggering Condition:** When you need to prove that segment ratios are equal (proportionality) or transfer an angle to another location.
    *    **Recommended Tool:** `draw_parallel_line`.
    *    **How to Use:** Draw a line from a key point that is parallel to an existing side of a triangle. This will create a smaller, similar triangle, allowing you to establish proportions between sides.

*    **Strategy: Segment "Cut and Paste" (Translation)**
    *    **Triggering Condition:** When you need to prove a relationship like `AB = CD + EF` or compare lengths of non-adjacent segments.
    *    **Recommended Tool:** `draw_translated_segment` or `draw_extended_line_by_length`.
    *    **How to Use:**
        1.  To "add" lengths, use ` draw_extended_line_by_length` to extend one segment by the length of another.
        2.  To "move" a segment for direct comparison, use `draw_translated_segment` to place it next to another segment. This helps to visualize equality or other relationships.

*    **Strategy: Finding Altitudes and Medians**
    *    **Triggering Condition:** When the problem mentions "altitude," "height," "area," or "midpoint."
    *    **Recommended Tools:** `draw_perpendicular_line` for altitudes; `draw_median_line` for medians.
    *    **How to Use:** These are direct constructions. Use `draw_perpendicular_line` to find the height of a triangle. Use `draw_median_line` to connect a vertex to

```
        the midpoint of the opposite side, which is useful in
        problems involving centroids or area division.

        *   **Strategy: Completing the Figure and Finding
        Intersections**
            *   **Triggering Condition:** When a problem's
        solution depends on the intersection of key lines like
        diagonals, altitudes, or angle bisectors which are not
        fully drawn.
            *   **Recommended Tools:** `draw_connecting_lines`
        followed by `draw_intersection_of_segments`.
            *   **How to Use:** First, use `draw_connecting_lines
        ` to draw the necessary segments (e.g., `["AC", "BD"]` to
         draw the diagonals of a quadrilateral). Then, if their
        intersection point is needed for further steps, use `
        draw_intersection_of_segments` to create and name that
        point.

# EXAMPLES:
**Example 1: Correct Usage**
<think> The problem asks to find the intersection of the
    diagonals in square ABCD. My first logical step is to
    construct these diagonals. The diagonals are the segments
     connecting opposite vertices, which are AC and BD. The `
    draw_connecting_lines` tool is designed for this. I will
    draw both segments at once. </think>
```json
{{
  "tool_name": "draw_connecting_lines",
  "parameters": {{
    "segment_names": ["AC", "BD"]
  }}
}}
```

---

**Example 2: Incorrect Usage (WHAT NOT TO DO)**
<think> The problem provides a triangle ABC. To start, I will
     draw the line segment AB to make sure it's part of my
    construction. </think>
```json
{{
  "tool_name": "draw_connecting_lines",
  "parameters": {{
    "segment_names": ["AB"]
  }}
}}
```
*This is wrong because line segment AB was already given as a
    side of the triangle. According to Rule #1, you must not
    draw lines that already exist.*

---

**Example 3: Correct Usage of `backtrack` for Self-Correction
    **
```

```
<think> My previous action was to draw a median from vertex C
    to side AB. However, after re-evaluating the problem, I
    realize my interpretation was wrong. The goal requires an
    **altitude**, which must be perpendicular, not a median.
    My last step was a logical error that led me down an
    incorrect path. Therefore, I must undo my last action
    before attempting the correct construction. I will use
    the `backtrack` tool to revert to the state just before
    my mistake (state 3). </think>
```json
{{
  "tool_name": "backtrack",
  "parameters": {{
    "to_state": 3
  }}
}}
```
*This demonstrates the correct use of self-correction. When
    you realize your previous step was logically flawed, use
    `backtrack` to undo it and clear the way for the correct
    approach.*
```

