# OpenReview forum: "GeoSketch: A Neural-Symbolic Approach to Geometric Multimodal Reasoning with Auxiliary Line Construction and Affine Transformation"
_ICLR.cc/2026/Conference — ICLR 2026 Conference Withdrawn Submission_

### Official Review · Reviewer_yfqc · 2025-10-28

**Soundness:** 3
**Presentation:** 3
**Contribution:** 2
**Rating:** 4
**Confidence:** 4

**Summary:**

This paper introduces GeoSketch, a neural-symbolic framework that addresses the limitation of static visual reasoning in MLLMs by enabling dynamic diagram manipulation through auxiliary line construction and affine transformations. The framework implements a perception-reasoning-action loop architecture and contributes a benchmark of 394 challenging geometry problems alongside a two-stage training methodology combining supervised fine-tuning and reinforcement learning. Experimental results demonstrate substantial improvements over static baseline approaches.

**Strengths:**

1. This paper accurately identifies a critical gap in current MLLMs for geometric problem solving: the inability to perform dynamic visuospatial manipulation.  By recasting geometric reasoning as an interactive perception-reasoning-action loop rather than static interpretation.
2. This paper proposes a well-designed three-tier architecture that effectively decouples visual perception, symbolic reasoning, and executable actions.
3. This paper contributes high-quality resources: the GeoSketch Benchmark with 394 curated problems requiring auxiliary construction or affine transformations, and a 2K fine-tuning dataset generated via neural-symbolic pipeline.

**Weaknesses:**

1. The experimental evaluation lacks comparisons on the complete MathVista geometry subset and PGPS9k with other geometric reasoning methods such as UniGeo, GeoX, etc.

2. Performance of Other Methods on GeoSketch Benchmark.

3. Could the authors provide comprehensive ablation studies?
- What is the individual contribution of each module (perception module, symbolic reasoning module, sketch action module)?
- Can similar performance be achieved with a simpler architecture?
- What is the detailed analysis of the action space design?  Which actions among the action types are most frequently used?  Are all action types necessary?

**Questions:**

1. The experimental evaluation lacks comparisons on the complete MathVista geometry subset and PGPS9k with other geometric reasoning methods such as UniGeo, GeoX, etc.

2. Performance of Other Methods on GeoSketch Benchmark.

3. Could the authors provide comprehensive ablation studies?
- What is the individual contribution of each module (perception module, symbolic reasoning module, sketch action module)?
- Can similar performance be achieved with a simpler architecture?
- What is the detailed analysis of the action space design?  Which actions among the action types are most frequently used?  Are all action types necessary?

---

### Official Review · Reviewer_pwt1 · 2025-10-28

**Soundness:** 3
**Presentation:** 2
**Contribution:** 2
**Rating:** 4
**Confidence:** 3

**Summary:**

The paper introduces GeoSketch, a neural-symbolic framework for geometric problem solving (GPS) that integrates perception, reasoning, and action in a closed loop. GeoSketch enables dynamic visual manipulation, such as auxiliary line construction and affine transformations, reflecting how humans iteratively reason through geometric proofs. GeoSketch is composed of three key modules: Perception Module, Neural-Symbolic Reasoning Module, Sketch Action Module. Combines supervised knowledge distillation and reinforcement learning with structured, symbolic rewards to achieve robustness and strategy generalization. The authors also introduce two datasets: GeoSketch Fine-Tuning Dataset – 2k verified trajectories emphasizing complex geometric reasoning, and GeoSketch Benchmark – 390 curated problems that require auxiliary constructions or affine transformations. In short, GeoSketch establishes a new paradigm of interactive, verifiable geometric reasoning, advancing MLLMs from static perception to dynamic cognitive manipulation grounded in symbolic logic.

**Strengths:**

1. The paper presents a novel reformulation of geometric problem solving as an interactive perception–reasoning–action loop, marking a conceptual shift from static multimodal reasoning to dynamic, executable visual interaction. The explicit action space for geometric manipulation (drawing, reflection, rotation, translation) extend the neuro-symbolic paradigm into a complex visuospatial reasoning domain.
2. In terms of dataset contributions, the authors construct a GeoSketch benchmark consisting of 390 problems that require dynamic manipulation, alongside a fine-tuning dataset of 2,000 problems annotated with solution trajectories. This effort addresses the gap in existing benchmarks, which lack evaluation for dynamic geometric reasoning, and provides a novel standard for assessing models in this domain.

**Weaknesses:**

1. The core contribution of the paper lies in the combination of a “dynamic closed-loop” mechanism with a neuro-symbolic architecture. However, the choice of techniques in some modules lacks sufficient justification in terms of novelty, making it susceptible to confusion with prior work. For instance, the perception module employs YOLO and U-Net to parse graphical elements, and reinforcement learning relies on the GRPO algorithm, both of which are established techniques, yet the paper does not clearly articulate any task-specific modifications for geometric reasoning.
2. While the idea of “dynamically modifying diagrams” represents a conceptual advancement, it partially overlaps with the “visual thinking chain plus drawing” paradigm proposed in Visual Sketchpad. The paper emphasizes only the difference of using “logical forms instead of code generation” but does not provide sufficient evidence that this difference yields substantial gains in problem-solving efficiency, for example, by comparing step redundancy or error rates under identical problem types.
3. The generalization evaluation is limited to "static geometry benchmarks" such as Geometry3K and GeoQA, and does not extend to more realistic scenarios, such as the geometry subsets within MathVista.

**Questions:**

1. The authors emphasize that GeoSketch’s "logical form-based graphical operation" differs from Visual Sketchpad (Hu et al., 2025)’s "code-based drawing." However, the paper only provides qualitative descriptions of this difference. Could you clarify: (1) What specific technical bottlenecks of code-based drawing does the logical form solve (e.g., code syntax errors, inefficient topology adjustment)? (2) Do you have quantitative data to prove this advantage (e.g., comparing the number of failed operations, step redundancy, or parameter adjustment time between the two methods on the same 50 complex geometry problems)?
2. The experiment only reports "final answer accuracy," but geometric reasoning often has "correct answers but flawed reasoning steps" (e.g., coincidentally getting the right result after misapplying a theorem or drawing unnecessary auxiliary lines). Would you kindly: (1) Provide the proportion of such "pseudo-correct" cases in your experimental results (e.g., how many of the correctly answered questions have logically flawed intermediate steps)? (2) Supplement the accuracy of theorem application and auxiliary line necessity in the intermediate steps (e.g., statistics on the rate of misusing theorems like congruence or similarity)?

---

### Official Review · Reviewer_6YWc · 2025-10-29

**Soundness:** 1
**Presentation:** 1
**Contribution:** 1
**Rating:** 2
**Confidence:** 5

**Summary:**

This paper proposes a neuro-symbiotic method to improve the geometric problem-solving abilities of large language models. Specifically, the authors proposed GeoSketch, which recasts geometric reasoning as a perception-reasoning-action loop. Specifically, the proposed method includes a perception module that abstracts diagrams into structured logic forms, a symbolic reasoning module that applies geometric theorems to decide the next deductive step, and a sketch action module that executes operations. The authors use SFT+RL to train a base model on a constructed dataset. Experiments show that the proposal can improve the performance.

**Strengths:**

The authors adopt the neuro-symbolic method to solve the geometric reasoning problem, and propose a perception–reasoning–action loop, which is technically sound.

**Weaknesses:**

1. The novelty is quite limited. There are too many works that follow a similar manner: SFT on some high-quality and RL fine-tuning to further improve the performance. No novelty and valuable insights are given.
2. The authors claim that they adopted a neuro-symbolic method; however, for the reasoning modules, they just adopt the LLM to conduct the reasoning. There is nothing symbolic.
3. For the experiments, the authors only conducted experiments on Geometry3K and GeoQA. More related datasets should be considered.
4. The authors propose that they compared with the learning paradigm "think with images", however, only VisualSketchpad is compared. Actually, there are many advanced works that tries to improve the "think with images" abilities of MLLMs. More related works should be included.

**Questions:**

As discussed above.

---

### Official Review · Reviewer_47Vc · 2025-10-31

**Soundness:** 2
**Presentation:** 2
**Contribution:** 2
**Rating:** 4
**Confidence:** 4

**Summary:**

The paper proposes **GeoSketch**, a neural–symbolic framework for geometric problem solving that recasts reasoning as an interactive *perception–reasoning–action* loop.
Unlike traditional static perception models, GeoSketch allows the agent to dynamically manipulate diagrams via auxiliary line construction and affine transformations.

The system consists of three components:
(1) **Perception**, which abstracts diagrams into symbolic forms;
(2) **Symbolic Reasoning**, which applies geometric theorems to guide deduction;
(3) **Sketch Action**, which executes visual operations to modify the diagram in a closed loop.

Training involves two stages:

* **Supervised Fine-Tuning (SFT)** on 2,000 teacher–student distilled trajectories;
* **Reinforcement Learning (RL)** with a hybrid “format + result” reward to improve robustness and planning.

Experiments on a new benchmark of 394 geometry problems show improved stepwise reasoning accuracy and problem-solving success compared to several multimodal LLM baselines.

**Strengths:**

1. **Integrating auxiliary-line construction into an agent loop** is conceptually novel and moves multimodal reasoning from static interpretation toward dynamic, executable interaction.
2. **Empirical performance** is reasonable: GeoSketch achieves consistent gains over strong baselines, indicating that neural–symbolic feedback and interactive sketch actions can enhance geometric reasoning.

**Weaknesses:**

1. **Limited methodological originality.**
   Most components (symbolic reasoning, auxiliary-line generation, SFT, RL) are established ideas combined within an agent architecture rather than newly invented techniques.

2. **Insufficient experimental analysis.**
   The paper lacks key ablations and detailed evidence to support its design choices.
   Specifically, the contributions of the *Sketch Action* module, the SFT stage, and the dual reward (format/result) remain unclear.
   While Section 3.3.1 briefly explains the teacher–student distillation process for SFT, important implementation details—such as teacher model identity, data filtering, and distribution—are missing.
   Similarly, although RL rewards are defined, there is no comparison showing why this combination is optimal.
   (More detailed issues are outlined in the Questions section.)

3. **Writing and formatting issues.**
   There is a minor unresolved cross-reference (“Section ??” in §3.3.1) and occasional awkward phrasing.
   These are not major flaws but should be corrected for clarity.

**Questions:**

1. **Ablation on the Sketch Action module.**
   Could the authors provide results for variants without visual actions or with textual-only auxiliary-line descriptions to quantify the effect of interactive diagram manipulation?

2. **Clarification of SFT data construction.**
   Section 3.3.1 describes a teacher–student distillation pipeline using 2,000 trajectories.
   Please specify which teacher MLLMs were used, how data quality was ensured, and whether any human curation or filtering was applied.

3. **Justification for the RL reward design.**
   The paper introduces “format” and “result” rewards but provides no evidence that this combination is superior or necessary.
   Can the authors include ablations or learning curves illustrating the effect of each component?

4. **Benchmark coverage and generalization.**
   Does the GeoSketch benchmark sufficiently cover diverse geometric reasoning types?
   How does the model generalize to unseen or out-of-distribution problems not requiring auxiliary constructions?

---

### Note · Authors · 2025-11-23

I have read and agree with the venue's withdrawal policy on behalf of myself and my co-authors.